# Efficient two-step chemoenzymatic conjugation of antibody fragments with reporter compounds by a specific thiol-PEG-amine Linker, HS-PEG-NH₂

Haruya Sato[ID][1]*, Yukiko Kataoka[2], Youichi Nishikawa[3], Daiki Okano[4], Mami Nagai[5], Yoshio Yamauchi[6], Masato Taoka[6]*, Katsuki Naitoh[7]

**1** Central Research Laboratories, Sysmex Corporation, Healthcare Science Hub Tokyo, Tokyo, Japan, **2** Bio-Diagnostic Reagent Technology Center, Technology Innovation, Sysmex Corporation, Healthcare Science Hub Tokyo, Tokyo, Japan, **3** Central Research Laboratories, Sysmex Corporation, Kobe, Hyogo, Japan, **4** Reagent Engineering, Sysmex Corporation, Kobe, Hyogo, Japan, **5** Reagent Engineering, Sysmex Corporation, Healthcare Science Hub Tokyo, Tokyo, Japan, **6** Department of Chemistry, Graduate School of Science, Tokyo Metropolitan University, Tokyo, Japan, **7** Strategic Technology Planning, Technology Strategy, Sysmex, Corporation Healthcare Science Hub Tokyo, Tokyo, Japan

\* Sato.Haruya@sysmex.co.jp (HS); mango@tmu.ac.jp (MT)

## Abstract

Chemoenzymatic conjugation of antibodies with reporter compounds offers broad applicability for detecting target antigens in the context of in vitro research and diagnostics. For conjugation, a bifunctional linker with a protected thiol and an amino group, serving as a transglutaminase substrate, is often employed. However, protective groups require an additional deprotection step during synthesis. To overcome this limitation, we selected a long-chain thiol-PEG-amine (HS-PEG) linker as the substrate. The HS-PEG linker exhibited minimal S–S bond dimerization in solution and was efficiently conjugated to Fab containing the transglutaminase-specific sequence tag (Q-tag), retaining the free state of the SH group. Using this SH group, a reporter compound containing two types of activated maleimides, Alexa488 or the red algae-derived fluorescent dye phycoerythrin, was conjugated to Fab. Both conjugates formed uniform structures in just two synthetic steps without compromising antigen-binding activity. Among the conjugates, phycoerythrin conjugated with multiple Fab molecules showed higher activity than that conjugated with fewer molecules in fluorescence enzyme-linked immunosorbent assays (ELISA) and flow cytometry. These results indicate that the chemoenzymatic approach using the HS-PEG linker and transglutaminase facilitates uniform Fab conjugation with reporter molecules for in vitro research and diagnostic applications. This method can expand the application of chemoenzymatic modification of antibody fragments by simplifying the conjugation process and reducing the formation of by-products.

**Data availability statement:** All relevant data are within the paper and its Supporting Information files.

**Funding:** The author(s) received no specific funding for this work.

**Competing interests:** The authors have declared that no competing interests exist.

**Abbreviations:** MTGase, microbial transglutaminase; HS-PEG linker, HS-PEG-NH2; Q-tag, transglutaminase recognition sequence tag; FCM, Flow cytometry; IVD, in vitro diagnostics; Fabp24, antibody fragment against HIV-1 p24 antigen; FabCD20, antibody fragment against CD20; PE, phycoerythrin; SEC, size exclusion chromatography; SPR, surface plasmon resonance.

## Introduction

Antibody conjugates with reporter molecules, such as fluorescent or chemiluminescent molecules, serve as powerful analytical tools for detecting target analytes in life science research and in vitro diagnostics (IVD) [1–4]. The conventional method for conjugating these reporter molecules to antibodies involves a chemical reaction using N-hydroxysuccinimide or maleimide derivatives, which target the amino group of lysine or the thiol group of cysteine residues of the antibody, respectively [5–8]. In these reactions, targeting specific residues among the multiple lysine and cysteine residues of the antibody is challenging, making it difficult to limit the stoichiometry to a certain range, leading to the synthesis of heterogeneous conjugates with low antigen-binding activity.

To achieve stoichiometric binding of reporter compounds to antibody molecules, most existing strategies require chemical modification of antibody carbohydrate-binding sites or the introduction of peptide tags, cysteine residues, or non-natural amino acids through molecular biological techniques [9,10]. Among these approaches, adding a "transglutaminase recognition sequence tag" (Q-tag) [11–13] to antibodies, followed by modification using MTGase, a protein glutamine γ-glutamyltransferase isolated from Streptomyces mobaraensis, has become a versatile tool for conjugate production owing to its high specificity and broad applicability [12,14–16]. In particular, a two-step chemical-enzymatic approach—where a linker containing a highly reactive residue, such as a thiol, azide, alkyne, or tetrazine group, is first attached to an antibody via MTGase, then reacted with a suitable reporter compound—is widely used for its high specificity and high conjugate yield [17–22]. Among these approaches, synthesizing derivatives through reactions with inexpensive thiol-containing linkers is preferred, as other functional groups are costlier and less practical.

Although the reaction of thiol groups with maleimide derivatives is useful for synthesizing conjugates using thiol groups [23–26], thiols must be free for this reaction to occur. However, free thiol groups are highly reactive and are typically blocked by protective groups such as acetylation or S–S bond reagents, which must be removed before use to prevent interference with other substances. The requirement for this deprotection step is a major limitation of an otherwise convenient reaction.

In this study, we used an HS-PEG linker to eliminate the need for protection and deprotection steps. The terminal thiol group of PEG was presumed to be stable enough in its free state to allow sufficient reaction time. Using this linker, we successfully constructed a stoichiometrically controlled conjugate of Q-tagged Fab with Alexa488 or phycoerythrin (PE) from red algae in a two-step process. Furthermore, we demonstrated that the Q-tagged Fab–PE conjugate is applicable for fluorescence (FL) ELISA and flow cytometry (FCM).

## Results

### Workflow and features of the conjugation method

This method synthesizes antibody-reporter conjugates by linking Fab and reporter compounds in two steps via an HS-PEG linker with amino and thiol groups at each end (Fig 1). In the first step, the amino group of the linker was conjugated to the

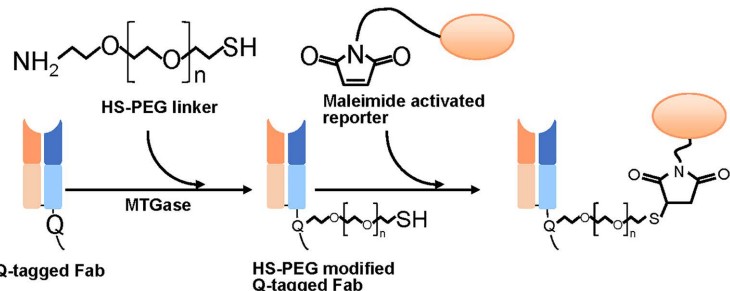

**Fig 1.  Schematic of our two-step chemoenzymatic modification of Q-tagged Fab with maleimide-activated reporter compounds.**

prepared Q-tagged Fab using MTGase forming HS-PEG-modified Fab. In the second step, the maleimide-functionalized reporter compound was coupled to the thiol group to complete conjugate construction. As MTGase and maleimide reactions are highly specific, this approach enables chemically controlled synthesis of antibody-reporter conjugates.

A key advantage of our method is that it eliminates the need to protect active thiol groups, which are commonly used in conjugate reactions [17,27]. This eliminates the protection and deprotection reaction steps from the synthetic procedure, thus simplifying conjugation. This strategy was achieved using an HS-PEG linker based on the speculation that the unique structure of this molecule creates spatial constraints, which prevent collisions between the thiol groups at the end of each molecule to inhibit the formation of S-S bonds. This is because: (i) this linker is sufficiently dispersed in water due to its high hydrophilicity [28], and (ii) its PEG portion has a trans-gauche-trans conformation that forms a long "cage-like" helical structure that restricts free movement in water [29,30]. Similar linkers have also been used in reactions with gold and silver particles without protective groups [31–35]. Therefore, we expected to achieve sufficient product yields without protection or deprotection.

## Characterization of the HS-PEG linkers by MS

To examine the molecular weight distribution of the HS-PEG2k linker, we performed direct infusion MS in the MTGase reaction buffer (Fig 2). The main signals in the MS spectrum corresponded to the calculated m/z values for HS-CH$_2$CH$_2$-(OCH$_2$CH$_2$)$_n$-NH$_2$ (Fig 2A). The spectra showed signals with degree of polymerization ranges of 39–55, with the most abundant signal, marked with an asterisk, observed at [M+H]$^+$ 2060.2, equal to the calculated value of [M+H]$^+$ at 2060.2 for the 45-mer of the HS-PEG2k linker. The other observed signals corresponded to Na adducts, not the dimer formed by the S–S bond. After an additional 180 min incubation, the spectrum showed that the dimerized HS-PEG linker signal (inverted black triangle) was only ~2% of the monomer (Fig 2B). This result indicates that most of the HS-PEG linkers retained their thiol groups after dissolution. This suggests that the MTGase-catalyzed first-step reaction can proceed efficiently without SH group protection due to rapid synthesis. We also analyzed the HS-PEG3.5k and HS-PEG5k linkers immediately after dissolution using direct infusion MS. The spectra showed signals with polymerization ranges of 64–95 and 86–124 (S1 Fig), a wider range than that of the HS-PEG2k linker. The major signal series in the MS spectra corresponds to those calculated from the structural formula, including H$^+$ and Na$^+$ adducts, and not to the dimer formed by the S–S bond. For subsequent experiments, we decided to use the HS-PEG2k linker reagent (n = 39–55) because it was more homogeneous and easier to handle.

## Proof-of-concept experiment for the method using the HS-PEG linker

We conducted experiments using an HS-PEG2k linker, Fabp24 as a typical Fab (Fig 3A), and the maleimide-activated Alexa488 as a reporter compound.

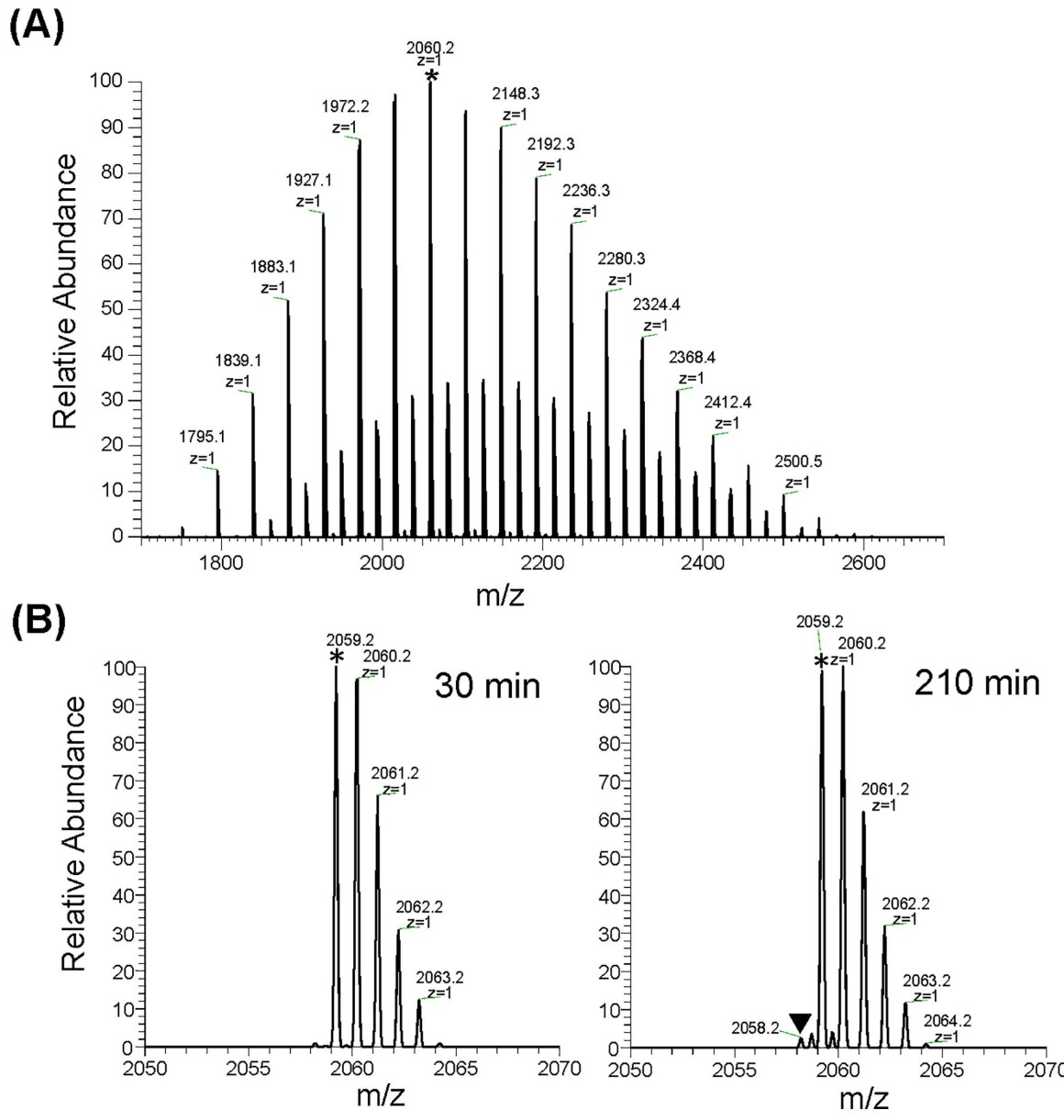

**Fig 2. Mass spectrometric characterization for the HS-PEG2k linker. (A)** Direct infusion MS spectra taken 30 min after solubilization of the HS-PEG2k linker reagent. Charge (z) and m/z are shown for some signals. Note that m/z for the major series of signals (every 44 Da) is consistent with m/z calculated from the structural formula of $HS\text{-}CH_2CH_2\text{-}(OCH_2CH_2)_n\text{-}NH_2$. The minor signal series was their Na adducts. **(B)** Infusion MS spectra at 30 (left) and 210 (right) min after solubilization of the HS-PEG2k linker, expanded around m/z 2059.2. Asterisks indicate the signal corresponding to monomer (n = 45) at m/z 2059.2 (z = 1). The triangle denotes the dimer signal (m/z 2058.2, z = 2) formed by an S–S bond..

In the first step, the amino terminus of the HS-PEG2k linker was reacted with Fabp24, catalyzed by MTGase. It was confirmed in advance that MTGase specifically introduced a linker molecule into this Q-tag using pentylamine-biotin as a probe (S2 Fig, S3 Fig and S1 Table). The reaction products were purified using protein G chromatography, concentrated by ultrafiltration, and monitored by size exclusion chromatography (SEC) (Fig 3B). In the presence of MTGase and the HS-PEG2k linker, the Fabp24 peak in the chromatogram appeared at an earlier elution time, suggesting successful

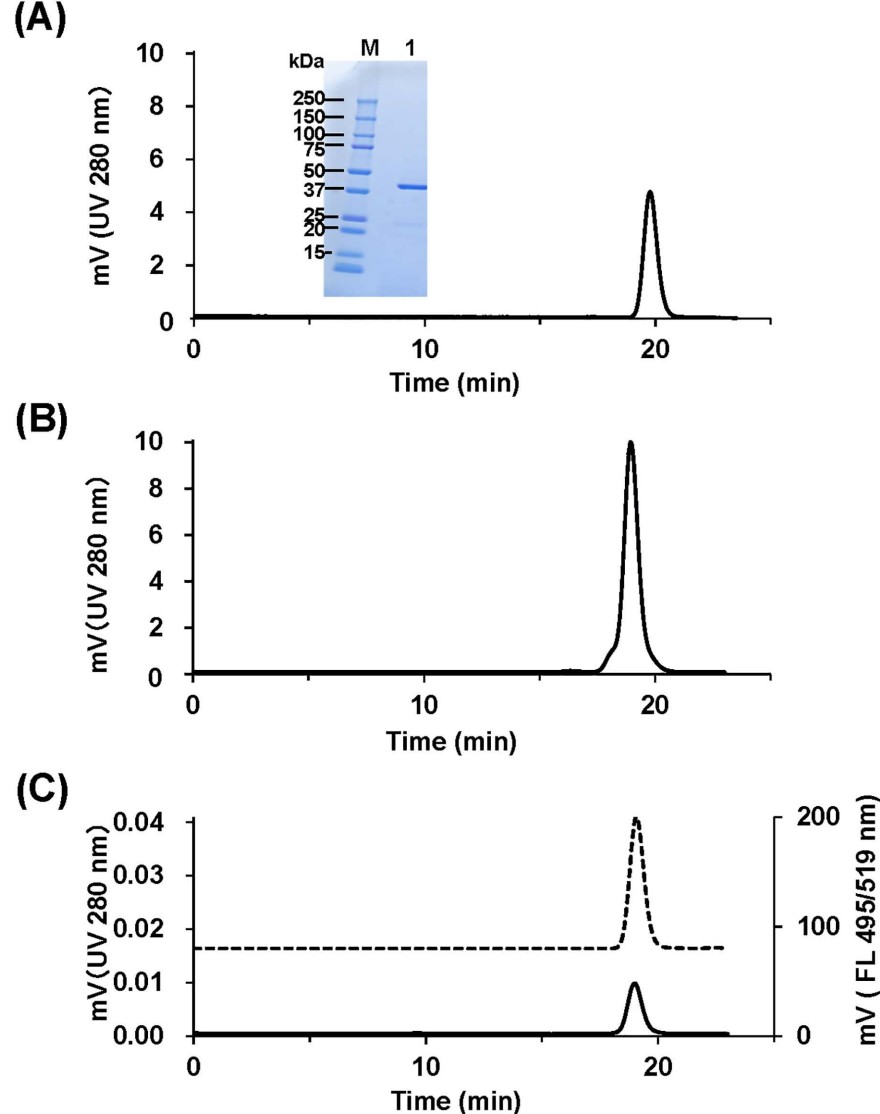

**Fig 3. SEC-HPLC profiles of Fabp24 and its conjugates.** (A) SEC and SDS-PAGE profile of Fabp24. Inset, non-reducing SDS-PAGE. Molecular weights of marker proteins are indicated. M, protein marker; 1, Fabp24. Fabp24 recognizes the HIV-1 p24 antigen, synthesized in-house (see Supplementary Information). Note that purified Fabp24 showed a single peak and band in SEC-HPLC and non-reducing SDS-PAGE. (B) SEC profile of HS-PEG2k-Fabp24. (C) SEC-UV and fluorescent profile of Alexa488-PEG2k-Fabp24. UV, solid line; FL, dotted line.

conjugation. The estimated reaction efficiency, based on the UV peak area, was high (>95%). Similar results were obtained while using the HS-PEG3.5k and HS-PEG5k linkers, which also reacted with Fabp24 at efficiencies comparable to that for HS-PEG2k (Fig S4).

In the second step, SEC-purified HS-PEG2k-Fabp24 was incubated with maleimide-activated Alexa488. Analytical SEC showed a single peak with both FL and UV absorption (Fig 3C), confirming a successful reaction. Although LC-MS analysis indicated an almost 100% reaction efficiency, the yield of Alexa488-modified HS-PEG2k-Fabp24 (Alexa488-PEG2k-Fabp24), based on the starting amount of Fabp24, was relatively low (0.1 mg, 33%), likely owing to conjugate loss during affinity chromatography and concentration steps.

## MS characterization of Alexa488-PEG2k-Fabp24

To characterize the conjugates, we performed LC-MS analyses of Fabp24, HS-PEG2k-Fabp24, and Alexa488-PEG2k-Fabp24 under reducing conditions (Fig 4). For unmodified Fabp24, multivalent MS signals of the heavy (H) and light (L) chains were observed (Fig 4, top panel). The deconvoluted average masses of the H and L chains of Fabp24 (24892.321 and 23588.539 Da, respectively) closely matched the theoretical values calculated from the sequences (24897.874 Da and 23593.337 Da, respectively). These results indicated that the H and L chains of Fabp24 have the designed sequences.

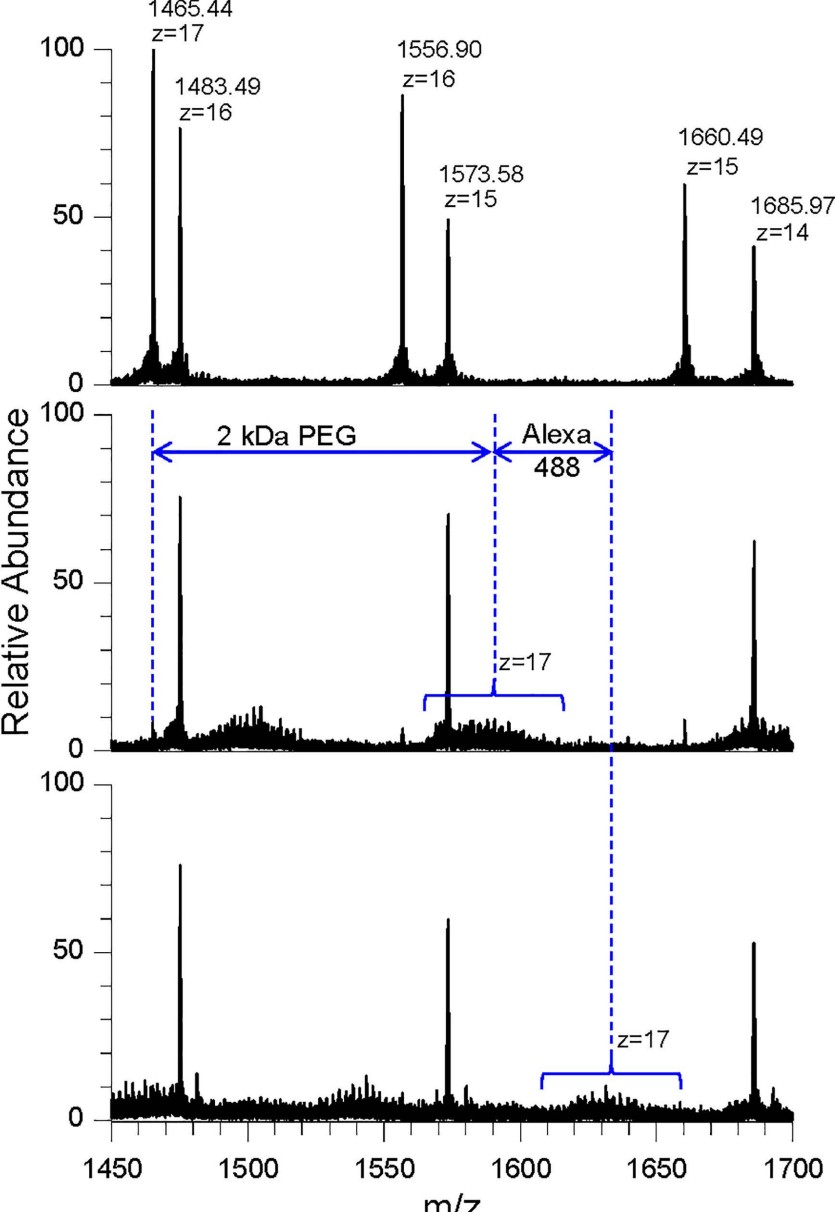

**Fig 4. LC-MS spectra of Fabp24 (top), HS-PEG2k-Fabp24 (middle), and Alexa488-PEG2k-Fabp24 (bottom).** Note that m/z 1465.44, corresponding to the heavy chain (z = 17), shifted to a higher molecular weight with the addition of the HS-PEG2k linker (middle) and Alexa488 (bottom), based on their molecular weight (blue arrows, dotted line, and parentheses). Similar shifts were observed for heavy chains at z = 16 and 15 (data not shown).

For HS-PEG2k-Fabp24, multiple signals were observed in Fig 4 (middle panel), likely representing H chains modified by HS-PEG2k with varying degrees of polymerization. The molecular mass differences between the H chains (Z = 17, Fig 4, top and middle panels) and 4(B)) closely matched the mass of the HS-PEG2k linker, suggesting site-specific conjugation of a single linker molecule to the H chain of Fabp24 by MTGase.

For Alexa488-PEG2k-Fabp24, the MS signals of the heavy chains in HS-PEG2k-Fabp24 showed an additional mass increase of approximately 700 Da (m/z 43, Z = 17, Fig 4, bottom panel), which was almost equal to that of maleimide-activated Alexa488 residue (697.7 Da). These results suggested that one molecule of maleimide-activated Alexa488 was incorporated into the H chain of HS-PEG2k-Fabp24, specifically in the SH group of PEG.

## HS-PEG linker with reporter compound does not affect Fab binding to target antigen

To compare the binding activities of Fabp24, HS-PEG2k-Fabp24, and Alexa488-PEG2k-modified Fabp24 against the HIV-1 p24 antigen, we conducted surface plasmon resonance (SPR) analyses for the purified samples (Table 1 and S5 Fig). Both modified Fabp24 showed KD values ($1.5 \times 10^{-9}$ M and $1.3 \times 10^{-9}$ M) similar to those of the unmodified Fabp24 ($1.4 \times 10^{-9}$ M) (Table 1). These results indicate that linker and reporter conjugation using this method did not reduce the antigen-binding activities of Fab.

## Applicability of the HS-PEG linker method for in vitro research uses

In practical applications of FL ELISA and FCM, PE is widely used as a highly fluorescent dye to detect low-level antigen expression [36]. Thus, we conjugated Fabp24 to PE via the HS-PEG linker, as described in the Methods section. Fig 5 shows SEC chromatograms of the reaction products between HS-PEG2k-Fabp24 and maleimide-activated PE. The starting materials decreased upon reaction, yielding at least three distinct conjugation products (products C, D, and E in Fig 5), all exhibiting UV absorption and PE-derived FL. The SEC-HPLC elution times and the UV/FL ratios of products

**Table 1. Dissociation constants of Fabp24, HS-PEG2k-Fabp24, and Alexa488-PEG2k-Fabp24 against HIV-1 p24 protein measured using SPR.**

| Fab and its derivatives | KD (M) |
|---|---|
| Alexa488-PEG2k-Fabp24 | $1.3 \times 10^{-9}$ |
| HS-PEG2k-Fabp24 | $1.5 \times 10^{-9}$ |
| Fabp24 | $1.4 \times 10^{-9}$ * |

*Average value of 2 independent data.

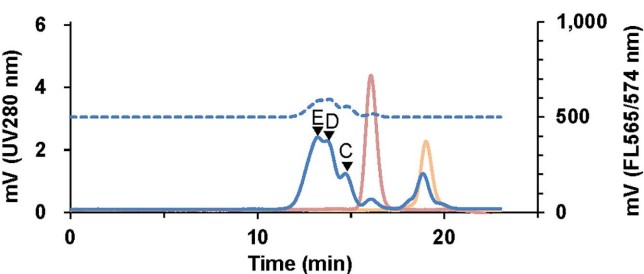

**Fig 5. SEC-HPLC chromatogram of the products of HS-PEG2k-Fabp24 reacted with maleimide-activated PE.** The reaction products of HS-PEG2k-Fabp24 and maleimide-activated PE were analyzed by SEC. The main products forming these peaks are marked with arrowheads C–E. UV, solid line; FL, dotted line; reaction product, blue; maleimide-activated PE, red; and HS-PEG2k-Fabp24, orange. The main products forming the peaks are marked with C–E and arrowheads.

C, D, and E increased in sequence, indicating that more HS-PEG2k-Fabp24 was conjugated per PE molecule. We concluded that products C, D, and E correspond to the mono-, di-, and trivalent HS-PEG2k-Fabp24 conjugates to a single PE molecule, respectively, as the UV/FL ratios of products C, D, and E were 1.0, 1.2, and 1.3, respectively, closely matching the calculated values (1.0, 1.1, and 1.2, respectively). Under the same reaction conditions, the quantitative ratios of products C, D, and E were consistently 1:2:3, as calculated from the peak areas monitored by UV absorption (Fig 5). We separated C, D, and E using preparative SEC (S6 Fig), and the resulting sample was subjected to FL ELISA against the HIV-1 p24 antigen (Fig 6A). At the same relative molar concentration, the FL signal increased in the order of C, D, and E, indicating that a higher number of Fab molecules per reporter molecule enhanced the affinity of the conjugate to HIV-1 p24.

This strategy was also applied to other Fabs. We synthesized FabCD20 by mimicking the rituximab sequence, which recognizes the extracellular domain of CD20 (Supplementary Results and S7 Fig). The yield of HS-PEG-modified FabCD20 conjugation to maleimide-activated PE was approximately 30% (data not shown). As with Fabp24, the reaction produced three conjugate species (C', D', and E') in a consistent 1:2:3 ratio, based on peak areas in the analytical SEC profile monitored by the UV absorption chromatogram (S7C Fig). After separation by preparative SEC, we assessed the binding activities against CD20-expressing Ramos cells using FCM (Fig 6B). At the same molar concentration, the mean FL intensity increased in the order of products C', D', and E', confirming that a higher number of Fab per reporter molecule induced a higher affinity of the conjugate to CD20.

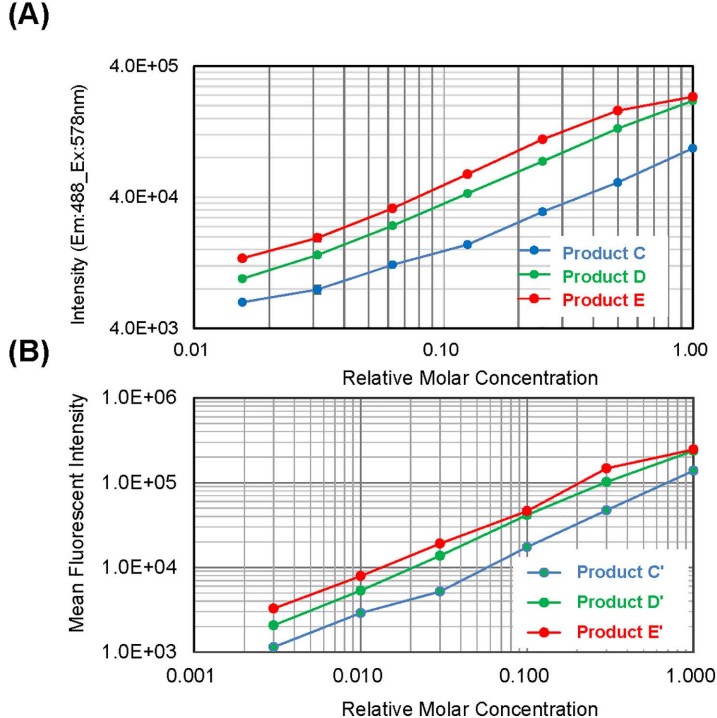

**Fig 6. Antigen detection ability of conjugates containing a few Fabs per PE molecule.** (A) ELISA plots for HIV-1 p24 protein using the product C-E of PE-PEG2k-Fabp24 as a probe. Each value represents the mean of two independent measurements. (B) FCM plots for CD20-expressing Ramos cells using the product C-E of PE-PEG2k-FabCD20 as a probe. Relative molar concentrations were adjusted by normalizing the FL intensity of each conjugate.

## Discussion

This method involves the synthesis of antibody-reporter conjugates in two steps using an HS-PEG linker. The linker was conjugated to a Q-tagged Fab using MTGase to create an HS-PEG-modified Fab. Next, the reporter compound, pre-attached to the maleimide group, was coupled to the thiol group, completing the conjugate. This method eliminates the need to protect active thiol groups, thereby simplifying the process. This method was validated using two Fabs (Fabp24 and FabCD20).

In the first step, the HS-PEG linker was used to bypass the need for thiol group protection during Fab modification (Figs 3 and 4). Additionally, a buffer containing a chelating agent was used during the MTGase reaction and subsequent purification to minimize metal-catalyzed intermolecular oxidation of the thiol group [37,38]. The reaction temperature was maintained at 5 °C to reduce intermolecular oxidation of the linker. Because the terminal amino group of PEG was a favorable substrate for MTGase [12,13,39], the HS-PEG linker was efficiently incorporated into the Q-tagged Fab, even at low temperatures. These reaction conditions are expected to facilitate the HS-PEG linker reaction as a monomer.

In the second step, commercially available fluorescent dyes (Fig 3) and PE (Fig 6) functioned as reporter compounds in the maleimide-activated form. PE, which has many lysine residues on its surface, is unlikely to induce structural changes owing to modification [40,41], allowing Fabs to bind without reducing FL intensity. This method should also be applicable to many other phycobiliproteins, including allophycocyanin [42,43], which are structurally similar to PE. In addition, we applied this method to maleimide-activated bovine intestinal alkaline phosphatase [44,45] without reducing its specific activity (data not shown). Using alkaline phosphatase and highly sensitive signal-generating transducers, such as AMPPD [46] and CDP-star [47], we believe that a highly sensitive method for detecting target antigens in biological samples can be constructed using a chemiluminescent enzyme immunosorbent assay [48,49].

The key point of our method is that it does not require either the protection or the deprotection of SH. It is more advantageous than conventional methods [9] because it eliminates (i) protection and deprotection steps, (ii) removal of by-products derived from each process, and (iii) use of additional reagents. In general, SH protection involves reaction by acetyl or S-S groups [17,27,50]. When protected by reacting with acetyl group, additional steps of deprotection with hydroxylamine and its removal are required before the reaction with maleimide derivatives [17]. When protected by reacting with S-S group, the use of $NH_2-CH_2-CH_2-S-S-CH_2-CH_2-NH_2$ as a linker [27] must be reduced before reacting with the maleimide, whereas this reduction breaks the interchain S-S bond, which is essential for antibody activity. In this case, the reconstitution of the interchain bond by re-oxidation is necessary, and a mismatched S-S bond is created as a side reaction. The method described in this paper can reduce the cost of producing conjugates by eliminating these troublesome steps.

As noted in the Results section, the yields of Alexa488-PEG2k-Fabp24 and PE-PEG2k-FabCD20 are around 30% and relatively low, respectively. The reason for these yields comes from the affinity chromatography with Protein G and concentration step by using ultrafiltration to remove the unreacted HS-PEG linker. Instead of this chromatography, we have obtained high yields by using cation exchange chromatography (over 80%, data not shown) for the upcoming scaled-up conjugation process.

Binding multiple Fabs to the reporter molecule via the HS-PEG linker resulted in enhanced antigen detection in the second step (Fig 6). There have been several reports of increased binding strength by immobilizing antibody fragments or antigens on particles or dendrimers with long-chain PEG linkers in a multivalent manner [51–54]. Forte et al. demonstrated that trivalent antibody fragments constructed using a three-branched PEG linker exhibited increased antigen avidity [55]. We consider that the increased intensity observed in the ELISA and FCM for the multivalent Fab conjugate with PE (Fig 6) was due to the increased avidity of the conjugate from multiple binding sites, as reported previously. The method reported here is highly reproducible in terms of the amount of multivalent Fab-conjugated PE synthesized (Figs 5 and S6C). By examining the quantitative ratio of each component in conjugate synthesis, high-avidity multivalent conjugates for IVD were reproducibly synthesized.

The maleimide-thiol conjugate used in this study was formed by Michael addition of the thiolates to the maleimide double bond [24–26]. However, these conjugates are susceptible to retro-Michael-type reactions leading to reporter compound loss during long-term storage in the presence of excess thiols, such as cysteine, glutathione, and serum albumin [38,56,57]. Hydrolyzing thiosuccinimide to form a succinamic acid thioether bond [57,38] enhances conjugate stability. Such long-term stability studies will be necessary for further conjugate development using the HS-PEG linkers.

Herein, we used transglutaminase to introduce an HS-PEG linker into the Q-tagged Fab. Transglutaminase catalyzes the acyl transfer reaction between the ε-amino group of lysine and the γ-carboxamide group of the glutamine residues in proteins, forming a stable isopeptide bond between the two substrates [58]. The recognition of a single substrate is not highly specific; thus, straight-chain amine derivatives, including lysine side chains, can be used [13,59–61]. The other substrate, the glutamine residue, and the surrounding amino acid sequence in the protein are highly specific [12,52,62,63], and even in high-molecular-weight proteins such as human IgG, only Q295 of the heavy chain is known to be a substrate [15,64]. A wide variety of glutamine-containing amino acid sequences are found in the variable regions of antibodies and antibody-related substances, making it difficult to demonstrate that only specific glutamine residues outside these regions are modified. The peptide mapping method shown here, using MS/MS scanning of product ions with m/z 329.20 [65], identified the glutamine modified by transglutaminase using pentylamine-biotin (S2 and S3 Figs and S1 Table). This MS/MS method will be highly useful not only for Fabs but also for various other proteins to confirm the site of transglutaminase modification with the HS-PEG linker before its introduction.

## Conclusion

This paper presents a chemoenzymatic method for conjugating antibodies with reporter molecules using an HS-PEG linker and transglutaminase. The method eliminates the need for thiol protection and enables efficient, uniform conjugation in just two steps. We believe that our study significantly contributes to bioconjugate chemistry because it simplifies the conjugation process, enhances antigen detection via formation of multivalent Fab-reporter constructs, and demonstrates broad applicability across different reporter compounds and proteins, including phycobiliproteins and enzyme-based detection systems.

## Materials and methods

### Chemistry

MTGase (Microbial Transglutaminase) was purchased from Activa KS-CT (Ajinomoto Corporation, Tokyo, Japan) and purified as described by Folk et al [66]. It has an activity of 29 U/mg, measured by the hydroxamate method, and allowed for use without differences with recombinant MTGase (30 U/mg) purchased from Zedira GmbH (Germany). HS-PEG-NH$_2$·HCl reagents with number-average molecular weights of 2,000, 3,500, and 5,000 (HS-PEG2k, HS-PEG3.5k, and HS-PEG5k linkers) were purchased from Sigma-Aldrich or Biopharma PEG (USA). N-ε-malemidocaproyl-oxysuccinimide ester (EMCS, E018) was purchased from Dojindo Laboratories (Japan). Maleimide-activated Alexa488 (product name: Alexa Fluor™ 488 C5 maleimide, MW: 720.7, Molecular formula: C30H25N4NaO12S2 CAS No: 500004-82-0) was purchased from Thermo Fisher Scientific (USA). Phycoerythrin (PE) from red algae was purchased from One Biotech (Taiwan). L-Cysteine HCl was purchased from Nacalai Tesque, Inc. (Japan). All other reagents and solvents were commercially available special-grade reagents and used without further purification.

### Mass spectrometry

Mass spectrometry (MS) was performed using a Thermo Q Exactive quadrupole-Orbitrap mass spectrometer (Thermo Fisher Scientific, USA) with a standard heated ESI source in positive mode. For LC-MS, an LC system (LC-20A, Shimadzu) equipped with a reversed-phase PLRP-1000 column (2 mm ID × 100 mm L, 8 μm particle size, PL Laboratory) was connected to the MS. MS spectra were analyzed using Xcalibur software (Thermo Fisher Scientific, USA).

For direct infusion MS analysis, HS-PEG linkers (3 mM final concentration) were dissolved in 20 mM Tris/HCl buffer containing 2 mM EDTA (pH 8.2) at room temperature and incubated for 30 or 210 min. Samples were diluted 100-fold with 30% acetonitrile containing 0.1% formic acid and immediately injected at a flow rate of 100 µL/min. The scan resolution was set to 30,000.

For LC-MS, samples were reduced with 200 equivalents of TCEP in 1M Tris/HCl buffer (pH 7.5) at 37 ℃ for 2 h and stored at −20 ℃. Subsequently, 50 µL of the sample was applied to LC-MS and eluted with a 10 min gradient of 10–60% acetonitrile in 0.1% formic acid at a flow rate of 100 µL/min. The eluate from the first 6 min was discarded as a salt-containing flow-through fraction using a three-way electric valve. The scan resolution was set to 140,000.

## Analytical and preparative SEC HPLC

For analytical SEC, a Chromaster™ (Hitachi High-Tech Science Corporation, Japan) LC system equipped with UV and FL detectors was used. An aliquot of the dialyzed sample was diluted 1:1 with Arg-SEC Mobile Phase buffer (Standard, Nacalai Tesque Inc., Japan) [67] and analyzed using a Superdex 200 Increase 10/300 GL column (10 mm ID × 300 mm L, 8.6 µm particle size, Cytiva, USA) at a flow rate of 0.8 mL/min at room temperature, as described in previous studies [67–70]. The concentration of the separated fraction was calculated based on the absorbance peak area at 280 nm, using recombinant human interleukin 6 [71] as a reference and applying the molar absorbance coefficient calculated using the method described by Pace et al [72].

Preparative SEC used the same system as analytical SEC, with 50 mM sodium phosphate containing 2 mM EDTA (pH 7.0) as the running buffer.

## MTGase-catalyzed modification of Q-tagged Fab with HS-PEG linker

The construction, expression, and purification of Q-tagged Fabs are detailed in the Supplementary Information. Given that the incorporation efficiency of long-chain PEG-NH$_2$ linkers into substrate proteins by MTGase is lower than that of low-molecular-weight pentylamine derivatives [16,62], the following modified conditions from a previously reported method [15] were applied. The MTGase-catalyzed modification of Fab with the HS-PEG linker was performed in a 20 mM Tris/HCl buffer (pH 8.2) containing 150 mM NaCl and 2 mM EDTA at 5 °C overnight with the following reagent concentrations: Q-tagged Fab, 10 µM; HS-PEG2k, 500 µM; MTGase, 0.1 U/mL. After the reaction, the conjugate was desalted and purified using a HiTrap Protein G HP column (1 mL, Cytiva, USA) according to the manufacturer's instructions and concentrated with Amicon ultra 10K (Merck, USA) to approximately 1/5 of the volume. The HS-PEG-modified Q-tagged Fab was stored at 5 °C until further use. The impurities and quantity of the conjugate were determined by analytical SEC.

## Conjugation of maleimide-activated Alexa488 with HS-PEG modified Fab

Maleimide-activated Alexa488 [73,74] was conjugated to HS-PEG-modified Q-tagged Fab in 50 mM sodium phosphate buffer (pH 7.0) containing 2 mM EDTA at 5 °C overnight under light shielding. The reagent concentrations were HS-PEG-modified Q-tagged Fab, 6 µM; maleimide-activated Alexa488, 120 µM. The reaction was quenched by incubating with L-cysteine·HCl (final concentration of 0.1 mM) at 5 °C for 1 h. The product was concentrated to approximately 1/2 of the volume using Amicon ultra 10K (Merck, USA) and separated by preparative SEC (see above). The eluted fraction with FL (excitation at 495 nm and emission at 519 nm) was collected and stored at 5 °C in a light-resistant container until further use.

## Measurement of dissociation constant between Fab derivatives and HIV-1 p24 antigen

Dissociation constants were determined by SPR (surface plasmon resonance) using a BIACORE™ T200 instrument (Cytiva, USA). Fab and its derivatives were immobilized on a CM5 sensor chip (Cytiva, USA) using a Human Fab Capture Kit (Cytiva) in HBS-EP+ buffer (Cytiva). Immobilization was performed with a contact time of 60 s at a flow rate of 30 µL/

min. Recombinant HIV-1 p24 protein (Prospec, Israel) at concentrations ranging from 2.5 to 80 nM was injected as an analyte with contact and dissociation times of 30 and 300 s, respectively, at the same flow rate. The interaction was analyzed using a 1:1 binding model with Biacore™ T200 evaluation software in one or two assays using five different analyte concentrations.

## Conjugation of PE and Fab through the HS-PEG linker

PE was dialyzed overnight at 5 °C in 50 mM sodium phosphate buffer (pH 7.0) containing 2 mM EDTA under light shielding and quantified by analytical SEC (see above). The dialyzed PE solution was activated at 37 °C for 1 h under light shielding with the following reagent concentrations: PE, 5 µM; N-ε-malemidocaproyl-oxysuccinimide ester 400 µM. The reaction mixture was desalted using a PD10 column (Cytiva, USA) with 50 mM sodium phosphate buffer (pH 7.0) containing 2 mM EDTA and concentrated to approximately 1/20 of the volume using an Amicon ultra 3 K column (Merck, USA). The concentration of maleimide-activated PE was measured using SEC.

Maleimide-activated PE was coupled with HS-PEG-modified Q-tagged Fab in 50 mM sodium phosphate buffer (pH 7.0) containing 2 mM EDTA at 5 °C overnight under light shielding with the following reagent concentrations: HS-PEG-modified Q-tagged Fab, 6 µM; maleimide-activated PE, 2 µM. The reaction solution was processed similarly to the Alexa488 conjugate, except that an Amicon Ultra 30K (Merck, USA) was used for concentration.

## Fluorescent ELISA measurement of the binding between PE-Fab conjugate and its antigen

The relative molar concentrations of PE-containing conjugates were normalized based on FL intensity (excitation at 488 nm and emission at 578 nm). The normalization was performed assuming that the PE of each conjugate had equivalent FL intensity. For PE-containing conjugates, dilution series were prepared in D-PBS (pH 7.4) and 0.1% BSA. A 96-well plate (Nunc™ 96-Well Polypropylene Sample Processing & Storage Microplates, Black, Thermo Scientific) was coated with an in-house anti-His tag antibody and reacted with HIV-1 p24 protein (HIV-1 p24 recombinant His tag, Prospec, Israel) for 1 h at room temperature. After washing the plates with saline containing 0.05% Tween 20, the PE-containing conjugate was added to the wells and allowed to stand for 1 h at room temperature to saturate the solid phase HIV-1 p24. The plate was washed again with saline containing 0.05% Tween 20, and the FL intensity (excitation at 488 nm and emission at 578 nm) was measured using a microplate reader (SpectraMax® GEMINI EM, Molecular Devices, USA).

## FCM analysis for detection of cellular antigen using PE-Fab conjugate

CD20-expressing Ramos cells (JCRB Cell Bank, Japan) were cultured at 37 °C in RPMI1640 medium supplemented with 10% fetal calf serum (FCS) in a humidified 5% CO2 atmosphere. The relative molar concentrations of PE-containing conjugates were normalized based on FL, as described above. The cells were harvested by centrifugation when the culture reached 80% confluence. The pelleted cells were resuspended in the wash buffer (D-PBS buffer (pH 7.4) containing 2% FCS and 2 mM EDTA), mixed with the Fab-containing conjugates in a 10 mL tube, incubated for 30 min at 5 °C to saturate cell surface-expressed CD20, and washed thrice with wash buffer. A dilution series of Fab-containing conjugates was prepared in a wash buffer. Cell FL was analyzed using a BD Accuri C6 Plus flow cytometer (Becton Dickinson Biosciences).

## Supporting information

**S1 Text. Contains the supplementary materials and methods, results, and references.**
(PDF)

**S1 Fig. Mass spectrometric characterization for HS-PEG3.5k and HS-PEG5k.** Charge (z) and m/z values are shown for certain signals. The m/z for the series of signals (every 44 Da) was consistent with the m/z value calculated using the structural formula $HS-CH_2CH_2-(OCH_2CH_2)_n-NH_2$. (A) The major signal series with charge and m/z were assigned to

[M+H+Na]$^{2+}$ ions. The minor signal series with a triangle is [M+2H]$^{2+}$. (B) The major signal series with charge and m/z were assigned to the [M+2Na]$^{2+}$ ions. The minor signal series with a triangle is [M+H+Na]$^{2+}$.
(PDF)

**S2 Fig. HIC-HPLC and electrophoretic analysis of the product of pentylamine-biotin-modified Fab(Q1) HIV1–25 by MTGase.** (A) HIC-HPLC profiles of Fabp24 modified with (solid line) or without (dotted line) pentylamine-biotin by MTGase. (B) Non-reducing SDS-PAGE and Western blot image of Fabp24 modified with or without pentylamine-biotin by MTGase. Non-reducing SDS-PAGE and subsequent Western blot analysis were performed following standard procedure. [11] Pentylamine-biotin was detected with HRP-conjugated streptavidin. M, molecular weight marker; 1, Fabp24; 2, Puri-fied product of MTGase-modified Fabp24 with pentylamine-biotin (peak B).
(PDF)

**S3 Fig. Base peak intensity and extracted ion chromatogram of tryptic peptides from Fabp24 and pentylamine-biotin-modified Fabp24.** Fabp24 conjugated, with or without pentylamine-biotin modification, was carbamoylmethylated, digested with trypsin, and analyzed using LC-MS and MS/MS. For (A) and (B), unmodified Fabp24 was used, and for (C), (D), and (E), pentylamine-biotin-modified Fabp24 was used. (A) and (C) show base peak intensity chromatograms; (B) and (D) display extracted ion chromatograms of the precursor ion at m/z 728.871 (±20 ppm). (E) presents the extracted ion chromatogram of the product ion at m/z 329.200 (±20 ppm). The asterisk in (D) denotes the target for MS/MS collision-induced dissociation analysis, as shown in Table 1. The extracted ion chromatogram in (E) at m/z 329.200 specifically depicts peptides modified by transglutaminase.
(PDF)

**S1 Table. Sequence-specific ions observed by MS/MS for pentylamine-biotin-labeled tryptic peptide from Fabp24 at m/z 728.780.**
(PDF)

**S4 Fig. SEC-HPLC chromatogram for the MTGase-catalyzed reaction products of Q-tagged Fab with the HS-PEG3.5k and HS-PEG5k.** Black dotted line, Fabp24 and MTGase reacted without the HS-PEG linker; black solid line, Fabp24 and MTGase with HS-PEG3.5k; blue line, Fabp24 and M TGase reacted with HS-PEG5k.
(PDF)

**S5 Fig. Sensorgrams of the SPR analyses of Fabp24 (A & C), HS-PEG2k-Fabp24 (D) and Alexa488-PEG-Fabp24 (B) against HIV-1 p24 protein.**
(PDF)

**S6 Fig. SEC-HPLC chromatogram of the products C (A), D (B), and E (C) of HS-PEG2k-Fabp24 reacted with maleimide-activated PE.** The reaction products of HS-PEG2k-Fabp24 and maleimide-activated PE were fractionated using preparative SEC and analyzed using analytical SEC. Products C–E correspond to those shown in Figure 5. UV: solid line; fluorescence: dotted line.
(PDF)

**S7 Fig. Analytical results of unmodified and modified FabCD20 by SEC-HPLC.** (A) SEC-HPLC chromatogram of FabCD20. (B) SEC-HPLC chromatogram of HS- PEG 2k -Fab CD20. (C) SEC-HPLC chromatograms of the coupling products of HS- PEG-modified FabCD20 with maleimide-activated PE . The reaction products of HS- PEG 2k -Fab CD20 and maleimide-activated PE were analyzed via analytical SEC. The main products forming the peaks are marked with C ' –E ' and arrowheads. UV, solid line; fluorescence, dotted line; reaction product, blue; maleimide-activated PE, sky blue ; HS- PEG 2k -Fab CD20 , orange . (D) SEC-HPLC chromatogram s of the products C ' , D ', and E ' of HS- PEG 2k -Fab CD20 reacted with maleimide-activated PE. The reaction products of HS- PEG 2k -Fab CD20 and maleimide-activated

PE were fractionated via the preparative SEC and analyzed by analytical SEC. Products C ' – E ' correspond to those of Figure S7(C) . UV, solid line; fluorescence, dotted line.
(PDF)

## Acknowledgments

We thank Yoneko Tobita and Yumiko Hosaka for the construction and purification of Q-tagged Fabs. We thank Masae Kurosaki for fruitful discussions on the development of conjugates for the IVD reagents. We thank Daisuke Ejima for their support and advice of SEC analysis. We also thank Kazuyuki Atarashi for their support and advice regarding FCM analysis. A patent application has been filed relating to this work.

## Author contributions

**Conceptualization:** Haruya Sato, Youichi Nishikawa, Masato Taoka, Katsuki Naitoh.

**Data curation:** Haruya Sato, Yukiko Kataoka, Mami Nagai, Yoshio Yamauchi, Masato Taoka, Katsuki Naitoh.

**Formal analysis:** Haruya Sato, Yukiko Kataoka, Daiki Okano, Mami Nagai, Yoshio Yamauchi, Masato Taoka, Katsuki Naitoh.

**Investigation:** Haruya Sato, Yukiko Kataoka, Youichi Nishikawa, Daiki Okano, Masato Taoka, Katsuki Naitoh.

**Methodology:** Haruya Sato, Yukiko Kataoka, Yoshio Yamauchi, Masato Taoka, Katsuki Naitoh.

**Project administration:** Haruya Sato, Katsuki Naitoh.

**Resources:** Haruya Sato, Katsuki Naitoh.

**Supervision:** Haruya Sato, Youichi Nishikawa, Yoshio Yamauchi, Masato Taoka, Katsuki Naitoh.

**Validation:** Haruya Sato, Yukiko Kataoka, Mami Nagai, Masato Taoka, Katsuki Naitoh.

**Visualization:** Haruya Sato, Yukiko Kataoka, Mami Nagai, Yoshio Yamauchi, Masato Taoka, Katsuki Naitoh.

**Writing – original draft:** Haruya Sato, Masato Taoka, Katsuki Naitoh.

**Writing – review & editing:** Haruya Sato, Youichi Nishikawa, Masato Taoka, Katsuki Naitoh.

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
