## [Decision Letter · Decision Letter 0]

22 Jul 2025

Dear Dr. Sato,

Thank you for submitting your manuscript to PLOS ONE. After careful consideration, we feel that it has merit but does not fully meet PLOS ONE’s publication criteria as it currently stands. Therefore, we invite you to submit a revised version of the manuscript that addresses the points raised during the review process.

We look forward to receiving your revised manuscript.

Kind regards,

Rui Tada, Ph.D.

Academic Editor

PLOS ONE

Journal Requirements:

2. We note that there is identifying data in the Supporting Information file <20250522_Supplement Information _Haruya_Sato.docx>. Due to the inclusion of these potentially identifying data, we have removed this file from your file inventory. Prior to sharing human research participant data, authors should consult with an ethics committee to ensure data are shared in accordance with participant consent and all applicable local laws.

-Location data

Please remove or anonymize all personal information (Name), ensure that the data shared are in accordance with participant consent, and re-upload a fully anonymized data set. Please note that spreadsheet columns with personal information must be removed and not hidden as all hidden columns will appear in the published file.

Reviewers' comments:

Reviewer's Responses to Questions

**Comments to the Author**

1. Is the manuscript technically sound, and do the data support the conclusions?

Reviewer #1: Yes

Reviewer #2: Yes

Reviewer #3: Partly

2. Has the statistical analysis been performed appropriately and rigorously?

Reviewer #1: Yes

Reviewer #2: N/A

Reviewer #3: N/A

3. Have the authors made all data underlying the findings in their manuscript fully available?

Reviewer #1: Yes

Reviewer #2: No

Reviewer #3: No

4. Is the manuscript presented in an intelligible fashion and written in standard English?

Reviewer #1: Yes

Reviewer #2: Yes

Reviewer #3: Yes

Reviewer #1: This study reports on the development of a novel methodology for conjugation of reporter molecules to antibody fragments under mild conditions, which is more straightforward compared to the previously developed approaches.

The study is well performed and the manuscript is well written. I only have minor questions:

In Figure 1, n= should be specified to what was used in this study.

Table 1, was the measurement only performed once? What is n= and SD=?

SI page 5 “LC-MS/MS analysis of tryptic digest of Fab. For LC-MS/MS analysis, 2 μL of 1 M Tris-HCl solution (pH 8.5) was added to 20 μL of 0.25 mg/mL Fab solution in milli-Q H2O containing 8 M urea (grade, maker).” - 8 M urea (grade, maker) should be added.

Reviewer #2: This paper describes the method for the preparation of Ab fragment-reporter conjugate by the two-step reaction: first step is the introduction of bifunctional linker to the Q tag attached to Ab via side chain amide bond. Subsequently, the thiol group of the linker was reacted with maleimide linked reporter. The advantage of this method is that the thiol group can be made free during the reaction with Ab, which realizes the direct coupling with the reporter without the deprotection step. Conceptually, the paper represents no surprising idea, as the use of Q-tag and transglutaminase has already reported and it is natural to think that the protection of thiol can be omitted in this reaction. However, the author successfully proved that the reaction indeed proceeded as suggested using several model proteins and this reviewer thinks that the paper may be suitable for publication with the following modifications.

1. SH-PEG-NH2 should be HS-PEG-NH2. In addition, in Fig. 1 PEG linker should be NH2....SH.

2. In page 3, line 8, making it difficult to control the stoichiometry-> The compound prepared in this paper is also heterogeneous, although at reporter part which does not affect the binding affinity. Still, it would be difficult to precisely control the stoichiometry to obtain the target conjugates. It is better to revise the expression.

3. Page 5, line 6: speculating that the steric hindrance... -> I do not understand the meaning. Please revise.

4. Page 8, line 5, page 13 line 3 from the bottom: likely owing to conjugate loss -> Please add the brief result of cation exchange data.

5. There is no experimental procedure for Table 1. Please add the errors of the KD value.

6. There are not MS data for the final conjugates. Please add them.

7. Do the authors intend to use this conjugate as a trimer? If so, it is better to optimize the yield of the trimer, as the purification by SEC is always necessary, which is not so efficient.

Reviewer #3: The authors have described a two-step chemoenzymatic method to generate antibody conjugate involving transglutaminase followed by a thio-maleimide Michael addition reaction. The key of this study is the demonstration of a long-chain thiol-PEG-amine (SH-PEG) linker is relatively stable than the short-chain thiol-PEG-amine during the transglutaminase mediated reactions, thus, to avoid the traditional thio group protection-deprotections procedure. Overall, the study provides a useful method for facilitating thio-maleimide Michael addition approach in antibody conjugate. However, several improvements should be made for before publication.

1. In Figure 2, the monomer is detected exclusively as a singly‑charged ion, while the dimer appears only as a doubly‑charged species. Although the dimer signal is minor (~2%, as stated by the authors), it would be more informative to include the mono‑charged form of the dimer as well, if it is present. This would help validate the assignment and improve the completeness of the mass spectrometric analysis.

2. Page 13, line 7, Fabp24 should not be purified by Protein G chromatography, unless it is engineered to be part of a fusion protein that binds to Protein G, which is not typical. It maybe a reason why the yield of Alexa488-PEG2k-Fabp24 is low (33%). Instead, Protein L should be applied.

3. Page 15, line 10: SPR spectra should be provided.

**Do you want your identity to be public for this peer review?** For information about this choice, including consent withdrawal, please see our Privacy Policy

Reviewer #1: No

Reviewer #2: No

Reviewer #3: No

---

## [Author Response · Author response to Decision Letter 1]

2 Sep 2025

Rebuttal letter to reviewer’s Comments

Thank you very much for reviewing our manuscript. We appreciate your useful comments and advice. We describe the replies to your comments and advice as blue letters.

In addition to the changes made in response to your comments, we have corrected the author's postal code (Manuscript, p1, line 3 from the bottom), which was incorrect. Furthermore, we have revised an erroneous description in Figure S7 (Supplementary Materials, p20, lines 1-2 and line 7). We appreciate your understanding in these corrections.

Reviewer #1: This study reports on the development of a novel methodology for conjugation of reporter molecules to antibody fragments under mild conditions, which is more straightforward compared to the previously developed approaches.

The study is well performed, and the manuscript is well written. I only have minor questions:

In Figure 1, n= should be specified to what was used in this study.

Thank you for pointing this out. Figure 1 is conceptual; thus we emphasized the number n (from 39 to 55) in another place in the text (p6, line 9).

Table 1, was the measurement only performed once? What is n= and SD

Thank you for your comments. The KD values for each sample were evaluated based on one or two assays (n= 1 or 2) using five different concentrations of analyte. To clarify this point, we have included the SPR experimental results and sensorgram figures (p7, lines; p15, Figure S5) in the Supplementary Information. We have also detailed the description of these results to include the equivalence of antigen-binding activity between unmodified and modified Fabp24, as follows (p7 from line 8 from the bottom to p8 line 2, in bold).

SPR analyses of Fabp24, HS-PEG2k-Fabp24, and Alexa488-PEG-Fabp24 against HIV-1 p24 protein.

Fab and its derivatives were immobilized on a sensor chip and injected at five different concentrations of recombinant HIV-1 p24 protein (Figure S5). As expected, HIV-1 p24 protein bound specifically to the Fabs. In contrast, no affinity for HIV-1 p24 protein was observed on sensor chips without the Fab (data not shown). The sensorgram profiles of HS-PEG2k-Fabp24 and Fabp24 (Figure S5A and S5B) and Alexa488-PEG2k-Fabp24 and Fabp24 (Figure S5C and S5D), each evaluated in a single batch of experiments, were highly similar, and their KDs were equivalent. Based on these results, the antigen-binding activity of modified Fabp24 and Fabp24 were considered equivalent.

Additionally, we added explanations to the footnote in Table 1 (p10) and the Materials and Methods section in the text (p20, line 6).

p10, Table 1 footnote: *Average value of 2 independent data (For Fabp24)

p20, line 5 (Bold): The interaction was analysed using a 1:1 binding model with Biacore™ T200 evaluation software in one or two assays using five different analyte concentrations.

SI page 5 “LC-MS/MS analysis of tryptic digest of Fab. For LC-MS/MS analysis, 2 μL of 1 M Tris-HCl solution (pH 8.5) was added to 20 μL of 0.25 mg/mL Fab solution in milli-Q H2O containing 8 M urea (grade, maker).” - 8 M urea (grade, maker) should be added.

We have added the information in the Supplement (p5, line 3).

Reviewer #2: This paper describes the method for the preparation of Ab fragment-reporter conjugate by the two-step reaction: first step is the introduction of bifunctional linker to the Q tag attached to Ab via side chain amide bond. Subsequently, the thiol group of the linker was reacted with maleimide linked reporter. The advantage of this method is that the thiol group can be made free during the reaction with Ab, which realizes the direct coupling with the reporter without the deprotection step. Conceptually, the paper represents no surprising idea, as the use of Q-tag and transglutaminase has already reported and it is natural to think that the protection of thiol can be omitted in this reaction. However, the author successfully proved that the reaction indeed proceeded as suggested using several model proteins and this reviewer thinks that the paper may be suitable for publication with the following modifications.

1. SH-PEG-NH2 should be HS-PEG-NH2. In addition, in Fig. 1 PEG linker should be NH2....SH.

Thank you for your advice. We have altered the nomenclature of SH-PEG and to HS-PEG in our revised manuscript. Further details are provided in the revised manuscript. In Figure 1, we also revised the description of linker’s structure as NH2-PEG-SH.

2. In page 3, line 8, making it difficult to control the stoichiometry-> The compound prepared in this paper is also heterogeneous, although at reporter part which does not affect the binding affinity. Still, it would be difficult to precisely control the stoichiometry to obtain the target conjugates. It is better to revise the expression.

We appreciate your advice. The expression has been revised in the text �p3, line 8-9�, as indicated in bold below.

In these reactions, targeting specific residues among the multiple lysine and cysteine residues of the antibody is challenging, making it difficult to limit the stoichiometry to a certain range, leading to the synthesis of heterogeneous conjugates with low antigen-binding activity.

3. Page 5, line 6: speculating that the steric hindrance... -> I do not understand the meaning. Please revise.

Thank you for your comments. We have revised the description in the text (p5, line 6-9), as indicated in bold below.

This strategy was achieved using an HS-PEG linker based on the speculation that the unique structure of this molecule creates spatial constraints, which prevent collisions between the thiol groups at the end of each molecule to inhibit the formation of S-S bonds.

4. Page 8, line 5, page 13 line 3 from the bottom: likely owing to conjugate loss -> Please add the brief result of cation exchange data.

Thank you for your indication. We have added the description about conjugate loss and brief results of cation exchange data in the revised text, as described below.

p8, line 6 (Bold):

likely owing to conjugate loss during affinity chromatography and concentration steps.

P14, line 4 from the bottom (Bold)

with Protein G and concentration step by using ultrafiltration to remove the unreacted HS-PEG linker.

p14, line 2 from the bottom (Bold):

Instead of this chromatography, we have obtained high yields by using cation exchange chromatography (over 80%, data not shown) for the upcoming scaled-up conjugation process.

5. There is no experimental procedure for Table 1. Please add the errors of the KD value.

Thank you for your comments. The KD values for each sample were evaluated based on one or two assays (n= 1 or 2) using five different concentrations of analyte. To clarify this point, we have included the SPR experimental results and sensorgram figures (p7, lines; p15, Figure S5) in the Supplementary Information. We have also detailed the description of these results to include the equivalence of antigen-binding activity between unmodified and modified Fabp24, as follows (p7 from line 8 from the bottom to p8 line 2, in bold).

SPR analyses of Fabp24, HS-PEG2k-Fabp24, and Alexa488-PEG-Fabp24 against HIV-1 p24 protein.

Fab and its derivatives were immobilized on a sensor chip and injected at five different concentrations of recombinant HIV-1 p24 protein (Figure S5). As expected, HIV-1 p24 protein bound specifically to the Fabs. In contrast, no affinity for HIV-1 p24 protein was observed on sensor chips without the Fab (data not shown). The sensorgram profiles of HS-PEG2k-Fabp24 and Fabp24 (Figure S5A and S5B) and Alexa488-PEG2k-Fabp24 and Fabp24 (Figure S5C and S5D), each evaluated in a single batch of experiments, were highly similar, and their KDs were equivalent. Based on these results, the antigen-binding activity of modified Fabp24 and Fabp24 were considered equivalent.

Additionally, we added explanations to the footnote in Table 1 (p10) and the Materials and Methods section in the text (p20, line 6).

p10, Table 1 footnote: *Average value of 2 independent data (For Fabp24)

p20, line 6 (Bold): The interaction was analysed using a 1:1 binding model with Biacore™ T200 evaluation software in one or two assays using five different analyte concentrations.

6. There are not MS data for the final conjugates. Please add them.

Thank you for your comment. As you pointed out, measuring the mass of the conjugate is very important. Unfortunately, these molecules are too large (>310 kDa) for our mass spectrometer, and we were unable to measure their masses.

7. Do the authors intend to use this conjugate as a trimer? If so, it is better to optimize the yield of the trimer, as the purification by SEC is always necessary, which is not so efficient.

We do not have a plan to use the (Fab-PEG)3-PE conjugate as a reagent for in vitro research or IVD reagents because its yield is too low. However, since the composition of HS-PEG-Fab conjugates with PE is highly reproducible under constant reaction conditions, we plan to use the total product fraction for our research and IVDs.

Reviewer #3: The authors have described a two-step chemoenzymatic method to generate antibody conjugate involving transglutaminase followed by a thio-maleimide Michael addition reaction. The key of this study is the demonstration of a long-chain thiol-PEG-amine (SH-PEG) linker is relatively stable than the short-chain thiol-PEG-amine during the transglutaminase mediated reactions, thus, to avoid the traditional thio group protection-deprotections procedure. Overall, the study provides a useful method for facilitating thio-maleimide Michael addition approach in antibody conjugate. However, several improvements should be made for before publication.

1. In Figure 2, the monomer is detected exclusively as a singly charged ion, while the dimer appears only as a doubly charged species. Although the dimer signal is minor (~2%, as stated by the authors), it would be more informative to include the mono charged form of the dimer as well, if it is present. This would help validate the assignment and improve the completeness of the mass spectrometric analysis.

Unfortunately, we were unable to detect the singly charged ion of the dimerized PEG molecule by MS analysis due to its high m/z. Thus, we estimated the amount of the molecule by the doubly charged ion.

2. Page 13, line 7, Fabp24 should not be purified by Protein G chromatography, unless it is engineered to be part of a fusion protein that binds to Protein G, which is not typical. It may be a reason why the yield of Alexa488-PEG2k-Fabp24 is low (33%). Instead, Protein L should be applied.

Thank you for your comment. We have made a preliminary comparison of the yield of HS-PEG-modified Fab using Protein G and Protein L chromatography, and the recoveries were similar. Since Protein G resin is cheaper than Protein L resin, we selected the Protein G column to purify the PEG conjugates and remove the unmodified HS-PEG linkers.

3. Page 15, line 10: SPR spectra should be provided.

We have included the SPR experimental results and sensorgram figures (p7, lines; p15, Figure S5) in the Supplementary Information. We have also detailed the description of these results to include the equivalence of antigen-binding activity between unmodified and modified Fabp24, as follows (p7 line 8-1 from the bottom to p8 line 2, in bold).

SPR analyses of Fabp24, HS-PEG2k-Fabp24, and Alexa488-PEG-Fabp24 against HIV-1 p24 protein.

Fab and its derivatives were immobilized on a sensor chip and injected at five different concentrations of recombinant HIV-1 p24 protein (Figure S5). As expected, HIV-1 p24 protein bound specifically to the Fabs. In contrast, no affinity for HIV-1 p24 protein was observed on sensor chips without the Fab (data not shown). The sensorgram profiles of HS-PEG2k-Fabp24 and Fabp24 (Figure S5A and S5B) and Alexa488-PEG2k-Fabp24 and Fabp24 (Figure S5C and S5D), each evaluated in a single batch of experiments, were highly similar, and their KDs were equivalent. Based on these results, the antigen-binding activity of modified Fabp24 and Fabp24 were considered equivalent.

---

## [Decision Letter · Decision Letter 1]

14 Sep 2025

Efficient Two-Step Chemoenzymatic Conjugation of Antibody Fragments with Reporter Compounds by a Specific thiol-PEG-amine Linker, HS-PEG-NH2

PONE-D-25-30206R1

Dear Dr. Sato,

We’re pleased to inform you that your manuscript has been judged scientifically suitable for publication and will be formally accepted for publication once it meets all outstanding technical requirements.

Kind regards,

Rui Tada, Ph.D.

Academic Editor

PLOS ONE

Additional Editor Comments (optional):

Reviewer #1:

Reviewer #2:

Reviewer #3:

Reviewers' comments:

Reviewer's Responses to Questions

**Comments to the Author**

Reviewer #1: All comments have been addressed

Reviewer #2: All comments have been addressed

Reviewer #3: All comments have been addressed

2. Is the manuscript technically sound, and do the data support the conclusions?

Reviewer #1: (No Response)

Reviewer #2: (No Response)

Reviewer #3: Yes

3. Has the statistical analysis been performed appropriately and rigorously?

Reviewer #1: (No Response)

Reviewer #2: (No Response)

Reviewer #3: Yes

4. Have the authors made all data underlying the findings in their manuscript fully available?

Reviewer #1: (No Response)

Reviewer #2: (No Response)

Reviewer #3: Yes

5. Is the manuscript presented in an intelligible fashion and written in standard English?

Reviewer #1: (No Response)

Reviewer #2: (No Response)

Reviewer #3: Yes

Reviewer #1: (No Response)

Reviewer #2: (No Response)

Reviewer #3: The authors have properly responded to my comments. I recommend the acceptance of the current manuscript.

**Do you want your identity to be public for this peer review?** For information about this choice, including consent withdrawal, please see our Privacy Policy

Reviewer #1: No

Reviewer #2: No

Reviewer #3: No

---

## [Editor Report · Acceptance letter]

PONE-D-25-30206R1

PLOS ONE

Dear Dr. Sato,

I'm pleased to inform you that your manuscript has been deemed suitable for publication in PLOS ONE. Congratulations! Your manuscript is now being handed over to our production team.

Kind regards,

on behalf of

Dr. Rui Tada

Academic Editor

PLOS ONE